# Evaluation of Exposure Doses of Elective Nodal Irradiation in Chemoradiotherapy for Advanced Esophageal Cancer

**DOI:** 10.3390/cancers15030860

**Published:** 2023-01-30

**Authors:** Hiroyasu Tamamura, Kenkei Hasatani, Sae Matsumoto, Satoko Asahi, Hitoshi Tatebe, Yoshitaka Sato, Keiichiro Matsusita, Yuji Tameshige, Yoshikazu Maeda, Makoto Sasaki, Shigeyuki Takamatsu, Kazutaka Yamamoto

**Affiliations:** 1Proton Therapy Center, Fukui Prefectural Hospital, 2-8-1 Yotsui, Fukui City 910-8526, Fukui Prefecture, Japan; 2Department of Gastroenterology, Fukui Prefectural Hospital, 2-8-1 Yotsui, Fukui City 910-8526, Fukui Prefecture, Japan; 3Department of Radiotherapy, Kanazawa University Hospital, Kanazawa 920-8641, Ishikawa, Japan

**Keywords:** esophageal cancer, chemoradiotherapy, elective nodal irradiation, proton beam therapy, pleural effusion, cardiac effusion

## Abstract

**Simple Summary:**

Surgery has shown that early-stage esophageal cancer is accompanied by extensive lymph node metastasis. Therefore, elective nodal irradiation is indispensable for chemoradiotherapy instead of surgery for the radical cure of esophageal cancer. In contrast, attention must be paid to adverse effects when applying radiation to a wide area. Adverse events are generally more intense with high-dose irradiation. Therefore, determining the appropriate dose for prophylactic irradiation to the elective nodal area is important. However, the appropriate dose has not been fixed. In this study, the therapeutic effects and adverse events of a group of 36 Gy elective nodal irradiation doses were similar to those of 40 Gy or higher. This study’s results indicate that preventing lymph node metastasis at lower doses is possible, thus suppressing adverse events.

**Abstract:**

We evaluated elective nodal irradiation (ENI) doses during radical chemoradiotherapy (CRT) for esophageal cancer (EC). A total of 79 patients (65 men and 14 women) aged 52–80 years with T1-3, N0-3, and M0 (including M1ly) who underwent CRT for EC during November 2012–September 2019 were eligible for this retrospective analysis. Patients were divided into two groups: the high-dose group (HG), including 38 patients who received ≥40 Gy as ENI; and the low-dose group (LG), including 41 patients who received <40 Gy. The median doses were 40.0 and 36.0 Gy in HG and LG, respectively. During the follow-up (median: 36.7 months), no lymph node recurrence was observed in the ENI field in all patients. Lymph node recurrence near the ENI field was observed in six patients. No significant differences were observed between the two groups in median overall survival, progression-free survival, and local control. Grade 3–4 acute and late adverse events were observed in five patients of HG and six patients of LG, respectively. No ulceration or stricture was observed in the ENI field on endoscopy examined with 58 Gy irradiation. In conclusion, an ENI dose of 36 Gy could be considered to control the elective nodes of EC.

## 1. Introduction

The incidence of esophageal cancer (EC) differs among regions. EC frequently occurs in central and eastern Asia [1], and eastern and southern Africa. EC is the sixth most common cause of cancer-related deaths worldwide and is detected at an advanced stage in most new cases [2]. Surgical resection is recommended as a treatment method for EC. Chemoradiotherapy (CRT) is the standard therapy for cases where surgery is impossible [3].

Considering that the lymphatics may be connected extensively and vertically along the esophageal wall is necessary while using a radiotherapeutic strategy for EC [4]. Therefore, EC tends to spread cranially and caudally at an early stage. Additionally, surgical outcomes have shown a high incidence of potential cellular metastases to a wide range of local lymph nodes [5]. This proves that the elective nodal irradiation (ENI) field cannot be excluded while determining the irradiation field for EC. ENI seems logical because a certain amount of irradiation is delivered to the lymph node fields with a risk of microscopic metastases; however, a large amount of irradiation delivered to large fields may cause more acute and late adverse events. Some studies on ENI have reported that 25–60% of patients experience acute toxicity of grade 3 or more, whereas 23–29% of patients experience late toxicity of grade 3 or more [6]. Hence, evaluating the irradiation fields and exposure doses of ENI is crucial; however, they remain unclear to date. Therefore, this clinical study aimed to evaluate ENI exposure doses by examining their therapeutic effects and adverse events.

## 2. Materials and Methods

### 2.1. Patients

In total, 131 patients with advanced EC who were judged inoperable or refused surgery at their request received CRT, including ENI, between November 2012 and September 2019, in our hospital. The diagnosis of EC was histologically confirmed with a biopsy before treatment in all patients. Endoscopy, esophagography, contrast-enhanced computerized tomography (CT), and positron emission tomography-CT (PET-CT) were performed for all patients, and the clinical stage of EC was assessed (International Union Against Cancer, version 8). Among the patients, those aged ≤ 80 years, with a performance status of 0–2, with T1 to 3, N0 to 3, M0 (including patients with distant metastases to the supraclavicular lymph nodes [M1ly]), and no history of receiving cancer treatment or chest radiation therapy (RT) were selected for this study. This study included patients who had received only chemotherapy before CRT. Subsequently, these patients were divided into two groups: the high-dose group (HG) which received 40 Gy or more as ENI with a cut-off value of 40 Gy; and the low-dose group (LG), which received less than 40 Gy.

The facility’s ethics committee relevant to the authors (institutional review board No. 13-68, No. 14-70) approved this study. Written consent was obtained from all participants of the study for using their data.

### 2.2. Radiotherapeutic Strategy

The tumor region was evaluated for all patients using endoscopy before RT-dedicated CT. The mucosal surface was evaluated using narrow-band imaging and chromoendoscopy, and the tumor margin was marked using short clips. The tumor was reflected as gross tumor volume (GTV primary [GTV_P_]) in the subsequent RT-dedicated CT image (slice thickness: 0.125 or 0.25 cm). Lymph nodes of size ≥ 0.5 cm in the minor axis of the CT and sites that showed higher fluorodeoxyglucose accumulation than the mediastinum on PET were considered positive for metastases (GTV nodes [GTV_N_]) [7]. The GTV included the primary focus [GTV_P_] and metastatic lymph nodes [GTV_N_]. Clinical target volume (CTV primary [CTV_P_]) was contoured by adding a 3.0 cm cranial margin and 0.5 cm right, left, dorsal, and ventral margins to the GTV_P_ [8]. CTV nodes (CTV_N_) were contoured by adding a 0.5–1.0 cm margin to the GTV_N_, and the ENI field included the CTV_N_ and elective irradiation field. No international definition of the ENI field has been established. Hence, in accordance with the fields used in the Radiation Therapy Oncology Group 85–01 trial and based on the proposals by Rice et al. at the authors’ institution, an ENI field between the supraclavicular region and the lesser curvature of the stomach, and corresponding to the surgical lymph node dissection area, was defined as a long-T field and that not including bilateral supraclavicular lymph nodes was defined as as a long-I field [9].

Meanwhile, in patients with cervical esophageal cancer and upper thoracic esophageal cancer, N0 to 1, a short-T field, including lymph node dissection fields at the time of surgery, proposed by Fujita et al. was used [10]. The short-T field area is the thoracic lymph node area, including the cervical lymph nodes, the supraclavicular lymph nodes, and the subcarinal lymph nodes.

Furthermore, the internal target volume was contoured by adding a 0.5–1.0 cm margin to the CTV to account for respiration and peristalsis. The planning target volume was also contoured by adding 0.5 cm to the CTV.

ENI was performed using 6 MV or 10 MV X-rays at a single exposure dose of 1.8 Gy or 2.0 Gy and was administered with breath control. Chemotherapy with 5-fluorouracil at 700 mg/m^2^ and cisplatin at 70 mg/m^2^ (FP70) was concomitantly administered to all patients. After ENI, localized areas (GTV_P_ and GTV_N_) were treated with intensity-modulated radiotherapy (IMRT) using 6 MV or 10 MV X-rays at 2.0 Gy per dose in combination with FP70, or proton therapy at 2.0 Gy per dose. The decision to use IMRT or proton therapy was made of the patient’s own free will. Multiple radiation oncologists discussed and set the exposure dose and irradiation field.

The dosimetric constraints for organs at risk met the following criteria by Li et al.: the exposure dose to the spinal cord was ≤45 Gy, the mean lung dose was ≤20 Gy, the percentage of lung volume receiving ≥20 Gy (V_20_) was <25%, V_30_ was ≤20%, and the mean exposure dose to the heart was <40 Gy, according to the tolerance dose TD5/5 [11,12]. Ray Station^®^ (RaySearch Laboratories AB, Stockholm, Sweden), XioN (Elekta, Mitsubishi Electric Corporation, Tokyo, Japan), and MIM Maestro^®^ (MIM Corp., Cleveland, OH, USA) were used to develop a therapeutic strategy and evaluate exposure doses.

Chemotherapy and supportive therapy were administered concomitantly during RT. Additionally, a physical examination, chest and abdominal CT, and esophageal endoscopy were performed at 3 month intervals for the first year after the treatment. Subsequently, an evaluation was performed every 3 or 6 months, and PET-CT was performed as required.

### 2.3. Evaluation of Toxicities

Acute and late adverse events were evaluated using National Cancer Institute Common Terminology Criteria for Adverse Events (CTCAE) version 5.0. Additionally, acute radiation mucositis was objectively evaluated using endoscopy when >50 Gy of irradiation was provided and the Fukui acute radiation esophagitis (FARE) grade by Hasatani et al. (Figure 1) [13].

### 2.4. Statistical Analysis

All statistical analyses were performed using SPSS version 27 (SPSS Inc., Chicago, IL, USA). Overall survival (OS) was defined as the period from the start of the treatment to death from any cause, censoring patients still alive at the last control visit. Complete response (CR) was defined as complete resolution of the disease by endoscopic evaluation. Otherwise (non-CR), a biopsy was performed. Patients were considered to have recurrence when a new tumor was observed in the ENI field, when tumor recurrence and regrowth were observed at the primary tumor site, and when endoscopic local biopsy results showed malignant cells cytologically. For the lymph node metastases, the onset of a new nodule in regions where no enlarged nodules were observed before irradiation was defined as regional lymph node recurrence (recurrence in the irradiation field). Newly enlarged lymph nodes outside of the irradiation field were defined as metastases to the lymph nodes outside of the irradiation field and were evaluated using PET-CT, as required. Progression-free survival (PFS) was defined as the period from the start of treatment to the date of the detection of tumor recurrence or the last follow-up date. The Kaplan–Meier method was used to estimate OS, PFS, and local control (LC). For comparisons between both groups, analyses were performed using the log-rank test (Mantel–Cox), chi-square test, and unpaired t-test, with the statistical significance set at *p* < 0.05.

## 3. Results

Among 131 patients who underwent CRT, 79 met the eligibility criteria. Table 1 shows the patient’s characteristics, including the 52 ineligible patients. The 79 eligible patients (38 patients with HG and 41 patients with LG) comprised 65 men and 14 women with a median age of 67 years (range, 49–80). Seventy-three cases (92.4%) had squamous cell carcinoma. In all treated patients, scheduled CRT was performed, and three-dimensional conformal RT (3D-CRT) or intensity-modulated RT (IMRT) were provided with 6 MV or 10 MV X-rays during ENI (Table 2). The median exposure dose was 40.0 Gy (40.0–48.0 Gy) in HG and 36.0 Gy (30.6–39.6 Gy) in LG, and the single exposure dose was 2.0 Gy in HG and 1.8 Gy in LG. In both groups, 20 doses of ENI were administered for 28 days, and the ENI treatment period was prolonged for ≥4 days in one patient in each (2.5%) of the groups. Large-area irradiation was delivered to the long-T or long-I ENI field in 68 patients (86.1%). Irradiation was performed with 3D-CRT in 76 patients (96.2%), and irradiation with IMRT was performed in three patients (3.8%). All three patients receiving IMRT irradiation had cervical esophageal cancer (Table 2).

### 3.1. Survival

During a median follow-up period of 36.7 months (range, 4.3–119.8), no new lymph node metastasis in the ENI field was noted in either group, but lymph node recurrence near the ENI field was detected in six patients (7.6%; Table 3). In patients treated with CRT, including ENI for advanced EC, the mean overall OS was 68.2 months (95% confidence interval [CI], 59.6–79.8); overall PFS, 64.7 months (95% CI, 52.2–77.2); and overall LC, 66.2 months (95% CI, 54.0–78.4). The 3-year OS rate was 65.0% (HG: 65.3%, LG: 64.9%), with no significant difference between the two groups (*p* = 0.848). The 3-year PFS rate was 59.0% (HG, 59.6%; LG, 58.2%), with no significant difference between the two groups (*p* = 0.749). After 3 years, the LC rate was 60.8% (HG, 62.8%; LG, 58.8%) with no significant difference between the two groups (*p* = 0.850; Figure 2).

### 3.2. Toxicities

Acute adverse events associated with CRT included grade 3 leukopenia in nine patients (four HG and five LG). For patients with grade 3 adverse events, the median boost dose was 30.0 Gy and the total irradiation dose was 60.0 Gy. One of these HG patients deteriorated to grade 4 as treatment progressed, and two patients had to discontinue treatment for 5 days. The median boost dose was 30.0 Gy, and grade 3 anemia was observed in six patients (two in HG and four in LG) who received a total irradiation dose of 66.0 Gy. Grade 3 thrombocytopenia was observed in one LG patient who was irradiated with a boost dose of 24.0 Gy and a total dose of 60.0 Gy (Table 4).

Based on CTCAE, grade 3 acute radiation esophagitis was observed in nine patients (two in HG and seven in LG). In 53 patients, the acute radiation mucositis under treatment was evaluated using endoscopy and classified based on the FARE grade when exposed to 58 Gy irradiation (Figure 1) [13]. In all patients, grade 2 or milder radiation mucositis was observed in the ENI field, and no stenosis or ulcer was observed. On the contrary, in the boosted irradiation field, grade 3 radiation mucositis was observed in one HG patient irradiated with a boost dose of 22.0 Gy. In addition, grade 4 radiation mucositis was observed in one LG patient who was irradiated with a boost dose of 37.0 Gy (Table 5).

No significant difference was observed between both groups in body weight changes at the start of the CRT treatment, after ENI, and at CRT completion. Additionally, the weight change after ENI was −3.9 kg for HG and −2.9 Kg for LG (Table 3). There were late adverse events, such as grade 3 pleural effusion in one patient in HG (total dose, 60.0 Gy) and grade 3 pericardial effusion in one patient in LG (total dose, 68.0 Gy). No grade 3 radiation pneumonitis was observed in either group (Table 4).

## 4. Discussion

Some studies, including meta-analyses, reported that prophylactic three-field lymph node dissection in radical surgery for EC could improve the survival rate [14]. Postoperative pathological examinations by Nishihira et al. showed cytological metastases to a wide range of lymph nodes in 31–56% of patients with T1b esophageal squamous cell carcinoma (SCC), 58–78% with T2, 74–81% with T3, and 83–100% with T4 [15]. Aggressive lymph node metastases should be suspected even in early-stage EC because the lymphatics might be connected extensively and vertically in the esophageal wall [4].

Although several studies have reported that no significant differences were observed in OS and PFS between CRT with ENI and CRT with involved field irradiation (IFI) only [16,17,18], IFI delivered only for reduced CTV prophylaxis is not considered to eliminate the micro-metastases of EC to lymph nodes. Zhu et al. retrospectively analyzed the clinical data on IMRT treatment for patients with 924 cervical or upper chest EC. They compared the clinical results of 272 patients with ENI and those of 652 patients with IFI and reported that the OS and PFS rates of the ENI group were significantly better than those of the IFI group [19]. In this study, the mean overall OS was 68.2 months (95% CI, 59.6–79.8), the overall PFS was 64.7 months (95% CI, 52.2–77.2), and the overall LC was 66.2 months (95% CI, 54.0–78.4). Furthermore, no lymph node recurrence was observed within the ENI region. Therefore, IFI delivered for reduced CTV prophylaxis does not eliminate the risk of lymph node recurrence.

ENI, which requires large-area irradiation, easily causes adverse events. According to a pooled analysis of 22 articles on CRT with ENI for EC conducted by Du et al., severe adverse events of acute hematotoxicity were observed in 15.7% of patients [20]. This study showed early CRT-associated adverse events of grade 3–4 leukopenia in nine patients (11.4%) (four in HG and five in LG). The consideration of hematotoxicity should be inevitable when administering ENI, especially in elderly patients with decreased marrow function and patients already being treated with anticancer drugs. Some studies indicated that the ENI group had a higher esophageal and pulmonary toxicity incidence [21,22]. Radiation esophagitis is usually evaluated based on CTCAE. The severity of acute esophagitis is graded according to the patient’s subjective sensations, such as pain and difficulty swallowing.

Here, acute radiation esophagitis was evaluated using the FARE grade described by Hasatani et al. [13]. Unlike the conventional method, the FARE grade is an evaluation method using an endoscope, and this method provides an objective evaluation of radiation mucositis. In other words, this method can be used to evaluate the grade of acute esophagitis in the ENI region and the grade of acute esophagitis in the boost region separately. In this study, the severity of radiation esophagitis was determined according to the endoscopic findings during 58 Gy irradiation (Figure 1). As a result, severe acute radiation esophagitis was observed in the boost field (grade 3 in one case and grade 4 in one case), but the FARE grade was evaluated as 0–2 in all patients in the ENI field, with no difference between HG and LG (Table 5).

In CRT for EC, attention should be paid to the late adverse effects on the heart and lungs. Kato et al. reported that the results of a phase II trial on CRT (JCOG9906) showed grade 3 or higher delayed adverse events of pericardial effusion in 16% of the patients, pleural effusion retention in 9%, radiation pneumonitis in 4%, and treatment-related death in four patients, during CRT [21]. Beukema et al. reported that pericardial effusion was observed in 48% of the patients [23] and other studies on CRT with ENI reported that 23–29% of patients experienced severe cardiopulmonary toxicity of grade 3 or more [24]. Grade 3 pleural effusion was observed in only one patient (1.3%) and grade 3 pericardial effusion in one patient (1.3%) in this study.

ENI with 3D-CRT was used in this study. Considering its effects on the heart, IMRT is considered better than 3D-CRT for ENI. However, when IMRT with a sharp dose gradient is delivered to large-area irradiation fields with costal and abdominal breathing and bowel peristalsis, attention should be paid to the reproducibility associated with movements that vary widely among individuals. Meanwhile, IMRT and proton beam therapy were applied as boost therapy for GTV_N_ and GTV_P_. A dose-volume histogram study showed that treatment with IMRT or proton beam therapy delivered a lower dose to the heart and lungs than the traditional X-ray treatment (3D-CRT), with significant benefits, especially for proton therapy [24]. The use of IMRT or proton beam therapy for boost irradiation may contribute to a significant reduction in late adverse events of the heart and lungs. In this study, grade 3–4 late adverse events occurred in two cases (2.5%). This is because IMRT and proton therapy were used as boost therapy in all cases, especially proton therapy in 53 cases (67.1%); this suggests that irradiation modalities, including proton therapy, may have contributed to the reduction of late adverse events.

Many studies administered irradiation at a fractionated ENI exposure dose of 1.8–2.0 Gy per day (total of 40–45 Gy) because irradiation has been traditionally administered at an exposure dose of 50.4 Gy in many facilities during radical CRT, and the ENI dose is the upper limit of the dose allowed into the spinal cord in CRT, including pre-operative CRT [6,25,26]. In contrast, Gignoux M et al. and Bosset JF et al. reported that preoperative irradiation with 33 Gy and 37 Gy, respectively, reduced lymph node metastasis in the irradiated area [27,28]. Furthermore, Suwinski et al. reported that a dose of 24 Gy could reduce metastases by 30–50% [29,30]. In this study, no lymph node recurrence was observed in ENI fields irradiated with 36 Gy (1.8 Gy x 20 fractions). Therefore, the results support the design of a randomized prospective study with ENI doses lower than 36 Gy.

This small, single-center study has several limitations. Patients consisted of both inoperable (45.6%) and operable (54.4%) patients who refused surgery of their own volition. Additionally, patients with stage 3 (41.7%) thoracic (79.7%) and esophageal SCC (92.4%) are the focus of the study. Although the effects and adverse events of CRT also depend on the anticancer drugs used in combination, this study only examined the results using FP therapy [31]. Larger randomized controlled trials are needed to clarify the appropriate dosage and ENI field.

## 5. Conclusions

This study showed no lymph node recurrence in the ENI field in HG and LG and no difference in LC, PFS, and OS. IMRT and proton beam therapy were used for the local regions (GTV_P_ and GTV_N_) boost therapy; however, there was no difference in the incidence of adverse events between the two groups. Therefore, CRT with ENI is useful, and therapy at an ENI exposure dose of 36 Gy (1.8 Gy × 20 fr) could be considered to control the elective nodes of EC.

## Figures and Tables

**Figure 1 cancers-15-00860-f001:**
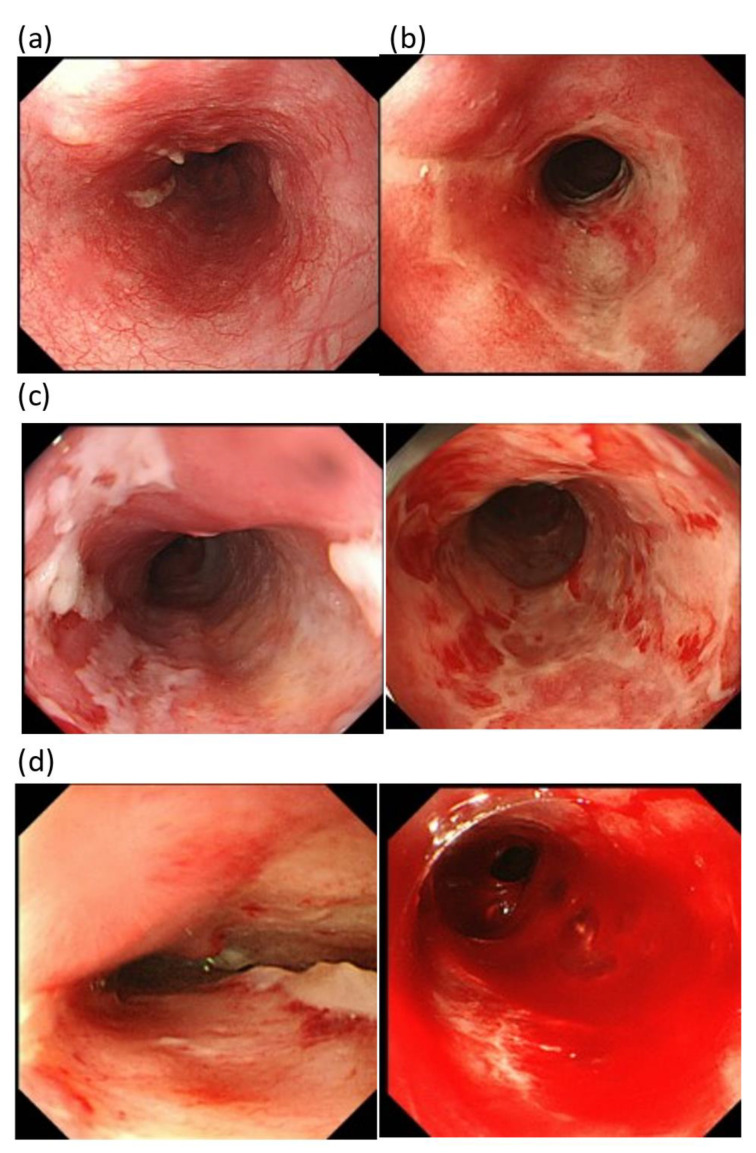
Radiation esophagitis grading using endoscopy (FARE grade). (**a**) Grade 1: mucosa with the erythema. (**b**) Grade 2: mucosa with erosion. (**c**) Grade 3: mucosa with a shallow ulcer and minor spontaneous bleeding. (**d**) Grade 4: mucosa with a deep ulcer and extensive spontaneous bleeding. This endoscopic photograph was kindly provided by the author, Dr. K. Hasatani.; Digestion. 101(4):366–374;2020.

**Figure 2 cancers-15-00860-f002:**
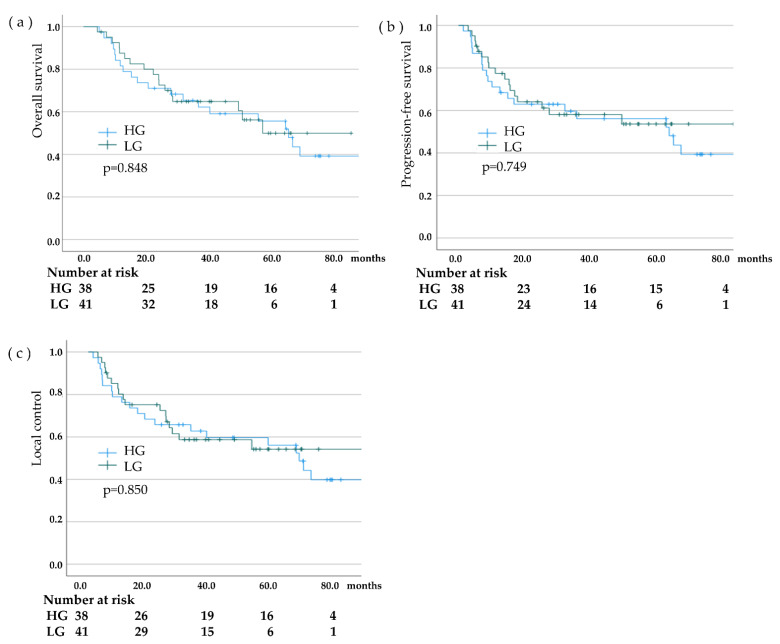
Comparison of the high-dose group (HG) and the low-dose group (LG) in overall survival (OS) rate, progression-free survival (PFS) rate, and local control rates. (**a**); Overall survival (OS) (**b**); Progression-free survival (PFS) (**c**); Local control (LC).

**Table 1 cancers-15-00860-t001:** Patient characteristics.

Characteristics	Patients	HG	LG	*p*-Value
Eligible patients, *n* (%)	79	38(48.1)	41(51.9)	
Follow-up time,	36.7	41.9	33	0.073
median (range), months	(4.3–119.8)	(4.9–119.8)	(4.3–84.1)	
Age, median age (range), years	67(49–80)	68(49–80)	66(52–80)	0.632
Gender, *n* (%)				0.554
Male	65(82.3)	31(81.6)	34(82.9)	
Female	14(17.7)	7(18.4)	7(17.1)	
Performance status, *n* (%)				0.755
0	45(57.0)	23(60.5)	22(53.7)	
1	26(32.9)	12(31.6)	14(34.1)	
2	8(10.1)	3(7.9)	5(12.2)	
Operability, *n* (%)				0.447
operable	36(45.6)	19(50.0)	17(41.5)	
inoperable	43(54.4)	19(50.0)	24(58.5)	
Histology, *n* (%)				0.771
Adenocarcinoma	6(7.6)	3(7.9)	3(7.3)	
Squamous cell carcinoma	73(92.4)	35(92.1)	38(92.7)	
T category UICC 8th, *n* (%)				0.564
T1	25(31.6)	14(36.8)	11(26.8)	
T2	12(15.2)	5(13.2)	7(17.1)	
T3	42(53.2)	19(50.0)	23(56.1)	
N category UICC 8th, *n* (%)				0.830
N0	19(24.1)	10(26.3)	9(22.0)	
N1	28(35.4)	12(31.6)	16(39.0)	
N2	29(36.7)	14(36.8)	15(36.6)	
N3	3(3.8)	2(5.3)	1(2.4)	
Stage UICC 8th, *n* (%)				0.438
I	19(24.1)	12(31.6)	7(17.1)	
II	16(20.3)	6(15.8)	10(24.4)	
III	33(41.7)	16(42.1)	17(41.4)	
IV	11(13.9)	4(10.5)	7(17.1)	
Tumor location, *n* (%)				0.735
Cervical	12(15.2)	7(18.4)	5(12.2)	
Thoracic	63(79.7)	29(76.3)	34(82.9)	
Upper/Median/Lower, *n*	10/30/23	5/14/10	5/16/13	
Abdominal	4(5.1)	2(5.3)	2(4.9)	

Abbreviations: HG = the high-dose group; LG = the low-dose group.

**Table 2 cancers-15-00860-t002:** Characteristics of the treatment status.

Characteristics	Patients	HG	LG	*p*-Value
Elective nodal irradiation (ENI)				
Median (range), Gy	39.6	40	36	<0.001
	(30.6–48.0)	(40.0–48.0)	(30.6–39.6)	
The single exposure dose, median (range), Gy	2.0(1.8–2.0)	2.0(1.8–2.0)	1.8(1.8–2.0)	<0.001
Irradiation fractions,	20(17–23)	20(20–23)	20(17–20)	0.175
Median (range)				
Irradiation period, (days) median (range)	28(22–36)	28(23–36)	28(22–34)	0.907
Large-area irradiation	68(86.1)	32(84.2)	36(87.8)	0.645
(Long T type + long I type), *n* (%)				
3DCRT irradiation method, *n* (%)	76(96.2)	37(97.4)	39(95.1)	0.602
Total dose, median (range), Gy	66	60	66	0.003
	(59.6–73.4)	(60.0–70.0)	(59.6–73.4)	
Irradiation combination				
(ENI + Boost Therapy)				
XT + PT, *n* (%)	53(67.1)	18(47.4)	35(85.4)	<0.001
XT + XT, *n* (%)	26(32.9)	20(52.6)	6(14.6)	
Chemotherapy				
Cisplatin and 5-fluorouracil, *n* (%)	79(100.0)	38(100.0)	41(100.0)	1.000
Ineligible patients	52			
Reasons for non-eligibility				
Over 81 years	19			
T4	12			
PS3-4	12			
ENI using PT	9			

Abbreviations: HG = the high-dose group; LG = the low-dose group; XT =X-ray therapy; PT = proton beam therapy.

**Table 3 cancers-15-00860-t003:** Changes in patient status with treatment (*n* = 79).

Changes in Patient Status	HG	LG	*p*-Value
Lymph node recurrence			
In the ENI field, *n* (%)	0(0.0)	0(0.0)	1.000
Near the ENI field, *n* (%)	4(10.5)	2(4.9)	0.344
Patient weight status			
Weight before treatment, kg	56.2	56.5	0.513
Weight change after ENI, kg (weight change, kg)	53.1(−3.1)	52.6(−3.9)	0.537
Weight after treatment, kg (weight change, kg)	52.3(−3.9)	53.6(−2.9)	0.437
The LC rate at 3 years, %	62.8	58.8	0.850
The PFS rate at 3 years, %	59.6	58.2	0.749
The OS rate at 3 years, %	65.3	64.9	0.848

Abbreviations: HG = the high-dose group; LG = the low-dose group; ENI =elective nodal irradiation; LC = local control; PFS = progression-free survival; OS =overall survival.

**Table 4 cancers-15-00860-t004:** Common Terminology Criteria for Adverse Events (CTCAE) Version5.0, grades for 79 patients.

	HG (*n* = 38)				LG (*n* = 41)				
	*n* (%)				*n* (%)				
CTCAE Grade	0 + 1	2	3	4	0 + 1	2	3	4	*p*-Value
Leukopenia									
Before treatment	38(100.0)	0(0.0)	0(0.0)	0(0.0)	39(95.1)	2(4.9)	0(0.0)	0(0.0)	0.616
After ENI	24(63.2)	10(26.3)	4(10.5)	0(0.0)	19(46.3)	17(41.5)	5(12.2)	0(0.0)	0.552
After treatment	26(68.4)	9(23.7)	2(5.3)	1(2.6)	25(61.0)	14(34.1)	2(4.9)	0(0.0)	0.703
Anemia									
Before treatment	37(97.4)	0(0.0)	1(2.6)	0(0.0)	41(100.0)	0(0.0)	0(0.0)	0(0.0)	0.694
After ENI	35(92.1)	2(5.3)	1(2.6)	0(0.0)	41(100.0)	0(0.0)	0(0.0)	0(0.0)	0.433
After treatment	31(81.6)	5(13.1)	2(5.3)	0(0.0)	34(82.9)	3(7.3)	4(9.8)	0(0.0)	0.289
Thrombocytopenia									
Before treatment	38(100.0)	0(0.0)	0(0.0)	0(0.0)	41(100.0)	0(0.0)	0(0.0)	0(0.0)	0.671
After ENI	37(97.4)	1(2.6)	0(0.0)	0(0.0)	41(100.0)	0(0.0)	0(0.0)	0(0.0)	0.328
After treatment	31(81.6)	7(18.4)	0(0.0)	0(0.0)	37(90.3)	3(7.3)	1(2.4)	0(0.0)	0.334
Pneumonitis	37(97.4)	1(2.6)	0(0.0)	0(0.0)	39(95.1)	2(4.9)	0(0.0)	0(0.0)	0.602
Pericardial effusion	38(100.0)	-	0(0.0)	0(0.0)	40(97.6)	-	1(2.4)	0(0.0)	0.333
Pleural effusion,	37(97.4)	0(0.0)	1(2.6)	0(0.0)	39(95.1)	2(4.9)	0(0.0)	0(0.0)	0.602
Esophagitis	19(50.0)	17(44.7)	2(5.3)	0(0.0)	13(31.7)	21(51.2)	7(17.1)	0(0.0)	0.098

Abbreviations: HG = the high-dose group; LG = the low-dose group; ENI =elective nodal irradiation; LC = local control; PFS = progression-free survival; OS = overall survival; FARE grade = the Fukui Acute Radiation Esophagitis grade.

**Table 5 cancers-15-00860-t005:** Evaluation of radiation esophagitis at the time of 58 Gy irradiation with ENI plus proton beam therapy (*n* = 53).

	HG (*n* = 18)		LG (*n* = 35)		
	*n* (%)		*n* (%)		
Grade	0–1	2–4 [2, 3, 4]	0–1	2–4 [2, 3, 4]	*p*-Value
The CTCAE grade	12(66.7)	6 [6,0,0 (33.3,0.0,0.0)]	11 (31.4)	24 [18,6,0 (51.4,17.2,0.0)]	0.014
The FARE grade	14(77.7)	4 [3,1,0 (16.7,5.6,0.0)]	26 (74.2)	9 [8,0,1 (22.9,0.0,2.9)]	0.780

Abbreviations: ENI = elective nodal irradiation; HG = the high-dose group; LG = the low-dose group; CTCAE = Common Terminology Criteria for Adverse Events (Version 5.0.); FARE grade = Fukui Acute Radiation Esophagitis grade.

## Data Availability

The data supporting the findings of this study are available within the article. The Supporting data are not publicly available due to their containing information that could compromise the privacy of research participants.

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
