# Peer review of "Evaluation of Exposure Doses of Elective Nodal Irradiation in Chemoradiotherapy for Advanced Esophageal Cancer"

_cancers, 2023, doi:10.3390/cancers15030860_

Round 1

Reviewer 1 Report

This paper deals with the ENI dose in the concurrent chemoradiotherapy using 5FU and CDDP (proton beam therapy is also used in some boosts) for esophageal cancer. This is a retrospective, single-center observational study in which the ENI dose was divided into two groups, the LD group (30.6–39.6 Gy, median 36 Gy) and the HD group (40.0–48.0 Gy, median 40 Gy), and the prognosis was compared.

Although it was a retrospective study, as for patient characteristics (follow-up time, age, gender, PS, histology, T, N, M, Stage, tumor location) and treatment factors (ENI period, rate of ENI large irradiation field, the rate of chemotherapy), there were no significant differences between the two groups. As a result, even at a low dose of 36 Gy to the ENI area, no recurrence occurred in the ENI area. Since there was no difference compared to the 40Gy high-dose group in several factors, overall survival, PFS, LCR, and acute and late adverse events in the ENI area, the authors conclude that the ENI dose of 36Gy may be sufficient. A characteristic feature of this paper is that in 53 patients in the proton therapy boost irradiation group, radiation esophagitis was evaluated not only by CTCAE but also by FARE grade using endoscopy.

Comments throughout

Among the results, it is better to state the results of OS, PFS, LC, and late adverse events in all patients. Then, as part of the discussion, the author will examine the position of the results of this research subject in terms of the literature. If the OS, PFS, LC, and late adverse events of all patients are equal to or higher than the conventional standard CRT results, the author should state that the results, comparing both HG and LG of ENI in this study, are worth considering although it is a retrospective study. Rather, the fact that acute myelosuppression, esophagitis, late pericardial effusion, pleural effusion, and radiation pneumonitis were lower than conventional standard treatment should also be mentioned. The author should emphasize that this minimally invasive is greatly involved of successfully combining high-precision radiation therapy such as IMRT and proton therapy.

Individual comments

1) The p-values for the comparison of HD and LD in Tables 2-5 are only partly shown in Table 2. All tables should display the calculated p-value.

2) Please describe the ENI and boost dose in cases with Grade 3-4 in the results.

3) In discussion, the author should consider the usefulness of FARE grade a little more.

Author Response

I would like to thank you for the many remarks you made on my incomplete paper.

I have made corrections in response to the doctor's remarks.

Following the corrected papers after the correction, I have attached what I have done and what I have corrected.

Reviewer 2 Report

The present trial studied the impact of two different doses of irradiation, ≥ 40 Gy vs ≤ 40Gy (median dose 40 Gy and 36 Gy, respectively), used while delivering elective nodal irradiation (ENI) for esophageal cancer (EC). No significant difference in terms of “in-field” nodal recurrence, OS, PFS, local control, G3-G4 acute and late toxicity was detected between  the two groups of patients. Authors concluded that in their experience a dose of 36 Gy for ENI was sufficient to control the elective nodes of EC.

Major comments

General comment.

Authors declared that the study was prospective but a clear study design is completely lacking. Probably the best method of comparing the effect of two different doses is a non inferiority study in which a pre-specified end point is clearly established in term of either oncological activity or toxicity. Usually this kind of study requires a number of patients much higher than that considered in this series but the conclusions are more reliable. Thus, I suggest to modify the statements “was” into  “could be” in the sentence “In conclusion, an ENI dose of 36 Gy was sufficient to control the elective nodes of EC”, line 37-38 of abstract.  The same in the conclusion section of the article, line 398-399: the verb “is considered to be enough” should be converted in “could be considered” to be enough or “seems to be enough to control... “

I also think that the sentence ”therefore, this results support the design of randomized prospective study with ENI doses lower than 36 Gy” (line 380-381 of discussion) should be added in the conclusions of abstract and article in order to highlight in advance the substantial exploratory nature of the study.

Methods

Radiation dose. There not seems to be a protocol planned dose. In effect, the Authors declared that “multiple radiation oncologists discussed and set the exposure dose and irradiation field” but they do not explain the criteria of choice.  Looking at the table 2 (and at the results section, line 164-165) prescription dose ranged from 30.6 Gy to 48 Gy. The Authors should explain the reason of this wide range, in particular the reason why it was decided to treat any single patient with “high” rather than “low dose”. This is a particularly crucial point because without a clear description of the choice criteria for prescription dose and field extension their experience is not reproducible.

Line 95-100. The Authors should better specify in the text when ENI long T- field and long-I field were chosen in relation to the tumor site. For instance, in case of N2 upper thoracic esophageal cancer, was the lesser curvature of stomach always included? In case of lower esophageal cancer was the long-I field always considered?

Line 99-100. The limits of the short-T field should be reported directly in the text without obliging the reader to look for the bibliographic reference.

Line 107. The Author should add that a radiation boost was delivered following ENI irradiation using X-rays or proton beams. Also in this case the Authors do not explain the criteria for choosing the dose (total dose ranged from 59.6 Gy and 73.4 Gy) and why 67.1% of patients were treated with protons and 32.9% with X-rays.

Line 142-143 Authors defined OS as the period of time from the start of treatment to the last observation date. Probably they mean: the period of from the start of treatment to death from any cause,  censoring patients still alive at the last control visit.

Line 142. “Complete response was evaluated using endoscopy”. The probably mean “complete disappearance of disease at endoscopic evaluation”. Were biopsies performed?

Line 146. Definition of recurrence. The sentence: ..”and when cancer cells were detected histologically or citologically… is not clear in the context of previous and following sentences. Is this referred to biopsies taken from esophagus? Lymph-nodes? Any other sites?

Minor comments

Abstract

Line 25: I suggest that the adjective “radical” before “chemoradiotherapy (CRT) for….should be added.

Methods

Line 105-118. The adverb “concomitantly” is used three times. The sentence included in the line 105-107 is perfectly clear and enough. The sentences in lines 108 and 118 should be cancelled.

Results

Outcome. Outcome data are good, in line or even better that those from best reports on preoperative treatment followed by surgery.

Toxicity. Please report the percentage of each toxicity data.

Discussion

I do not have substantial comments relative to the  discussion section. Probably, it would be better to make it a little shorter and better focused on the equal results in terms of both toxicity and efficacy between the HD and LD groups. The opportunity of administering a lower dose, as reported in lines 370-377 of discussion, should be more emphasized and, if possible, better explored in the literature.

Could the use of protons have had a positive impact on the efficacy other than on toxicity as hypothesized in line 369?

Line 306-309. The meaning of the sentence is not clear to me.

Author Response

(The authors gave the same response as above.)

Reviewer 3 Report

Dr. Tamamura and his colleagues compared OS, PFS, LC, and toxicities between high dose irradiation and low dose irradiation group. They concluded these outcomes showed similar, and ENI dose of 36Gy was sufficient.  

(Major problems)

・Median radiation dose of HG and LG were 40Gy and 36Gy, respectively, Even if these were significantly different between the groups, the difference seems tiny. Is this such a small difference so important? It is no wonder that since the difference is small., outcomes of HG and LG groups resulted in similar.

・Is this study a retrospective study or RCT? And how did the authors divide HG and LG group? The authors should show the design of this study.

・The authors should describe the treatment strategy for EC. In this study, cases of early and advanced EC were included, However, especially for advanced EC, surgery is a promising treatment. Why patients in this study received CRT?

(Minor problems)

・I could not understand the definition of local recurrence. Is that a matter of only primary lesion? Since the author described that no recurrence was observed in the ENI field,

・The authors should show p-value in Table 3.

・Discussion is too long.

Author Response

(The authors gave the same response as above.)

Round 2

Reviewer 1 Report

It seems to have been fixed properly.

Reviewer 2 Report

The manuscript is acceptable in this review version.

Reviewer 3 Report

Thank you for replying my question and suggestion. The manuscript seems to be adequately revised, and valuable to publish.